# Feasibility of Ultrasound-Guided, Peripherally Inserted Central Catheter Placement at the Bedside in a Communicable-Disease Isolation Unit

**DOI:** 10.3390/jpm13050863

**Published:** 2023-05-20

**Authors:** Kyoung Won Yoon, Wongook Wi, Moon Suk Choi, Eunmi Gil, Chi-Min Park, Keesang Yoo

**Affiliations:** 1Division of Critical Care, Department of Surgery, Chung-Ang University Gwangmyeong Hospital, Gwangmyeong 14353, Republic of Korea; kywon.yoon@cauhs.or.kr; 2Department of Anesthesiology and Pain Medicine, Chung-Ang University Gwangmyeong Hospital, Gwangmyeong 14353, Republic of Korea; hestia.w@gmail.com; 3Department of Surgery, Inha University Hospital, Inha University School of Medicine, Incheon 22332, Republic of Korea; 4Department of Critical Care Medicine, Samsung Medical Center, Sungkyunkwan University School of Medicine, Seoul 06351, Republic of Korea; 5Division of Acute Care Surgery, Department of Surgery, Samsung Medical Center, Sungkyunkwan University School of Medicine, 81 Irwon-ro, Gangnam-gu, Seoul 06351, Republic of Korea

**Keywords:** peripheral catheterization, central catheterization, vascular access device, ultrasonography, COVID-19

## Abstract

Background: Previous studies have investigated the safety of peripherally inserted central catheters (PICCs) in the intensive care unit (ICU). However, it remains uncertain whether PICC placement can be successfully carried out in settings with limited resources and a challenging environment for procedures, such as communicable-disease isolation units (CDIUs). Methods: This study investigated the safety of PICCs in patients admitted to CDIUs. These researchers used a handheld portable ultrasound device (PUD) to guide venous access and confirmed catheter-tip location with electrocardiography (ECG) or portable chest radiography. Results: Among 74 patients, the basilic vein and the right arm were the most common access site and location, respectively. The incidence of malposition was significantly higher with chest radiography compared to ECG (52.4% vs. 2.0%, *p* < 0.001). Conclusions: Using a handheld PUD to place PICCs at the bedside and confirming the tip location with ECG is a feasible option for CDIU patients.

## 1. Introduction

Peripherally inserted central catheters (PICCs) are used in various clinical settings, such as nutritional support, chemotherapy and critical care [1,2,3]. Although central venous catheter (CVC) access is a common practice in intensive care units (ICUs), a previous study demonstrated that this procedure is associated with several complications, including central line–associated bloodstream infection (CLABSI). In ICU settings, utilization of PICCs is deemed appropriate when their expected usage duration is 15 days or longer. On the other hand, if peripheral access is required for a period of 5 days or less, peripheral IV devices, such as midline catheters, are considered suitable [3]. Furthermore, a recent study demonstrated that insertion of PICCs at the bedside in the ICU was both safe and highly successful, with an overall success rate of 98.1%. The superiority of the current technique is evident when compared to the previous blind method or traditional PICC placement, which is performed by an interventional radiologist in the interventional radiology suite under fluoroscopic guidance [4,5]. Kim et al. [5], demonstrated that ultrasound-guided PICC placement performed by intensivists, not radiologists, may be safe and feasible, with incidence of CLABSI infection not significantly different between groups. PICC placement in patients with severe respiratory distress was recommended during the coronavirus disease-19 (COVID-19) pandemic, and intracavitary electrocardiography (ECG) and echocardiography were reported to save resources and increase safety under such circumstances. The benefits associated with utilizing PICCs in COVID-19 patient scenarios are the following: the absence of pleuropulmonary complications during insertion, making them suitable for patients experiencing severe respiratory distress. In pronated patients, PICCs offer easy management of the exit site. For patients who are anticoagulated, PICCs pose less risk of bleeding. They also cause minimal to no interference with respiratory management in patients receiving noninvasive ventilation and continuous positive airway pressure. Finally, PICCs are beneficial for tracheostomized patients [6]. Despite recommendations for CVC or PICC implantation during the COVID-19 pandemic, there is a lack of real-world studies on PICC implantation in highly restrictive settings, such as communicable-disease isolation units (CDIUs). These units are typically small and urgently installed during pandemics and may not have all of the necessary equipment of a full-equipped ICU. Furthermore, the nurse-to-bed ratio also shows a difference between CDIUs and ICUs. In February 2020, the nurse-to-patient ratio reached its highest point, with a peak ratio of 1.57. However, in the following months, the ratio declined significantly, dropping below 0.6 in both April and May of 2020, according to a Singaporean study [7,8]. As far as we know, there has not been any research conducted on the procedure of inserting PICCs in CDIUs that were established during the COVID-19 pandemic. Consequently, the objective of this study was to examine the practicality of conducting PICC insertions on patients admitted to CDIUs, which typically have fewer resources compared to ICUs.

## 2. Materials and Methods

### 2.1. Study Design and Population

This retrospective study was conducted between January 2020 and December 2021 at a university-affiliated tertiary referral hospital, and it received approval from the institutional review board (IRB No. 2022-07-186). As this study was retrospective in nature, the need for patients’ informed consent was waived. This study included patients, aged 19 or more years, who were admitted to the CDIU for COVID-19 treatment and underwent PICC insertion. The CDIU of our institute operates as a single-cell room unit, separated from the other ICU wards (Figure 1). Due to the challenges posed by wearing personal protective equipment (PPE) and having relatively fewer nursing staff than in the ICU, accessing the intravenous route by conventional methods was difficult with patients admitted to the CDIU. As a result, there was a high demand for PICC-placement procedures among these patients. COVID-19 diagnosis was confirmed through detection of severe acute respiratory syndrome coronavirus 2 (SARS-CoV-2) using a reverse transcription polymerase chain reaction (RT-PCR). The RT-PCR test was conducted using the PowerChek™ 2019-nCoV real-time RT-PCR kit manufactured by Kogene Biotech, based in Seoul, South Korea.

### 2.2. Procedure

Before starting of the procedure, the patient’s medical history, including chronic renal disease, was reviewed to check for the presence of an arteriovenous fistula or the need for left-arm vein preservation for fistula creation. In accordance with the guidelines of the Department of Infection Control of our institution, hand sanitizing was performed using a 4% chlorhexidine gluconate soap solution. During the procedure, healthcare personnel followed strict infection control protocols by wearing appropriate PPE. This included sterile surgical gowns, double gloves, facial shields, goggles, surgical caps and N95 masks. Some individuals also used powered air-purifying respirators (PAPRs) as an added precaution. To maintain a high level of cleanliness, the skin was meticulously cleansed using a solution comprising 2% chlorhexidine gluconate mixed with 70% isopropyl alcohol. Additionally, throughout the entire procedure, maximum sterile barrier precautions were consistently maintained. Ultrasonographic examination was performed using the linear array transducer of a handheld portable ultrasound device (PUD, Vscan Extend™, GE Healthcare, Milwaukee, WI, USA). Sterile transducer covers and sterile gel were used in all ultrasonographic examinations and procedures. Local anesthesia was performed using a subcutaneous injection of 1–3 mL of 1% lidocaine. In the CDIU, a 5-French (Fr) triple-lumen PowerPICC™ catheter (Bard Access Systems, Salt Lake City, UT, USA) or a 5-Fr dual-lumen Turbo-Ject^®^ Power-Injectable PICC (Cook, Bloomington, MN, USA) was used. Under the guidance of ultrasound imaging using a linear probe, vein localization and venipuncture procedures were performed after the application of a tourniquet. The basilic vein was the preferred site for insertion, although the brachial and cephalic veins were also utilized when appropriate. The insertion process involved sequentially introducing a guidewire with a modified Seldinger technique, a peel-away sheath and a peripherally inserted central catheter (PICC) using the over-the-wire technique. To ensure proper placement, the location of the PICC tip was confirmed either with electrocardiography (ECG) or by obtaining a portable chest radiograph. Giant P-waves were observed when the catheter tip was located near the right atrium (Figure 2). In patients with arrhythmia or those requiring continuous ECG monitoring, the location of the catheter tip was confirmed using portable chest radiography [9,10,11]. ECG was preferred over radiography to prevent contamination from additional radiological equipment in the procedure field of the CDIU. Therefore, if ECG could not confirm the catheter-tip location, radiography was preformed after the end of the procedure.

### 2.3. Data Collection, Endpoints and Statistical Analysis

After obtaining of approval from the IRB, patient data were extracted from electronic medical records (EMRs). Various variables were collected, including age, sex, height, weight, length of hospital stay, admission date and time to the CDIU, procedure date and time, diagnosis, medical history, complications, type of catheter, accessed arm and vein, method of catheter tip confirmation and utilization of point-of-care ultrasound. The primary focus of this study was to investigate complications associated with PICC insertion. The primary endpoint involved comparing the number of patients in whom catheter-tip position was verified using ECG versus those confirmed using chest radiography. The secondary endpoint was the occurrence of catheter malposition in confirming the catheter-tip position using both ECG and chest-radiography methods.

Statistical analyses were performed using R (R Core Team2022, R: A language and environment for statistical computing; R Foundation for Statistical Computing, Vienna, Austria). For continuous data, we used the median and the interquartile range (IQR) to describe the distribution. Categorical data are presented in the form of numbers and percentages.

## 3. Results

### 3.1. Characteristics of Patients

During the study period, 79 patients, admitted to the CDIU for COVID-19 management and who underwent PICC placement, were enrolled. However, after careful review of their EMRs, five patients were excluded because their CVCs were not PICCs; the Arrow^®^ 7 Fr three-lumen catheter (Teleflex^®^, Wayne, PA, USA) was used in three patients, and the Power-Trialysis™ triple-lumen hemodialysis catheter (Bard Access Systems, Salt Lake City, UT, USA) was used in two. As a result, the final analysis included a total of 74 patients. The baseline characteristics of our cohort are summarized in Table 1. Local anesthesia was not performed in seven patients because they were deeply sedated with analgesics and sedatives. In the total cohort, more than 40% (*n* = 31) of venous access procedures were performed within 24 h after CDIU admission. ECG was performed (*n* = 51) in nearly 70% of the patients to confirm the location of the catheter tip. After the procedure, the physician unintentionally omitted confirming the catheter-tip location using chest radiography in one patient. Venipuncture procedures were guided by PUDs in 73 (98.65%) patients. The vein was not identified by the PUD in one patient, necessitating the use of a conventional high-end ultrasound device. When the PICC-placement procedure was performed, most of the patients had high oxygen-demand statuses and received high-flow nasal cannulas (HFNCs, 45.95%) or mechanical ventilation (36.49%).

### 3.2. Procedural Outcomes

Regarding complications, it was observed that four patients developed hematomas, and there were five instances where a break in aseptic technique occurred during the insertion procedure. In order to confirm the correct placement of the catheter tip, electrocardiography (ECG) was performed on 52 patients, while portable radiography was carried out on 21 patients. Out of these, a total of twelve cases of malposition were reported, with eleven occurring in the radiography group and one in the ECG group. Immediate reinsertion of the catheter was performed in these cases, which accounted for 52.38% in the radiography group compared to 1.96% in the ECG group (*p* < 0.001). It is important to interpret this result cautiously, as radiography was used as a sequential method following PICC placement when ECG was not suitable, particularly in patients with arrhythmia. However, ECG was the preferred method because use of portable radiography in the CDIU involves heavy consumption of time and human resources. The most common reasons for PICC placement were the need for non-peripheral-compatible infusion medications and the expected requirement of intravenous access for more than two weeks. In nine patients, PICC placement was planned before the extracorporeal membrane oxygenation procedure to save the intrajugular veins. None of the patients developed deep-vein thrombosis (DVT) at the site of PICC insertion until discharge.

## 4. Discussion

The outbreaks of COVID-19, severe acute respiratory syndrome (SARS) and Middle East respiratory syndrome (MERS) have caused significant changes in healthcare strategies. Even after identifying SARS-CoV-2, we had no prior knowledge that it would exhibit significant differences compared to previous coronaviruses, such as SARS-CoV or Middle East respiratory syndrome coronavirus (MERS-CoV). Our initial response to COVID-19 followed a similar approach to our actions against SARS and MERS. Compared to SARS, the overall fatality rate of COVID-19 is lower. Nevertheless, the failure of our containment efforts can be attributed to the fact that COVID-19 patients shed the virus during the early stages, even when displaying mild symptoms [12,13]. During the COVID-19 pandemic, public health guidelines [13,14] emphasized the importance of maintaining physical distance and using PPE to reduce the spread of the virus. However, healthcare providers have encountered numerous obstacles when it comes to upholding the quality of medical services for patients who require isolation due to the pandemic. This is primarily due to the limited availability of medical resources and capabilities within emergency departments (EDs) and ICUs. Effective management protocols are crucial for timely planning and resource allocation, as they can reduce deviations from standard care and improve patient outcomes. Well-being of staff and prevention of staff burnout are also important factors that greatly impact teamwork effectiveness and can have an influence on patient outcomes [15]. Peck [13] proposed establishment of additional care centers, referred to as “isolation-and-care facilities”, as part of the nation’s preparedness strategy. Furthermore, a contingency plan should be devised to bolster critical-care capacity. The influx of COVID-19 patients surpassed the capacities of both isolation beds and ICU beds during the pandemic, necessitating the urgent setup of CDIUs. These CDIUs are often hastily installed during pandemic situations and are typically smaller in size, lacking essential equipment in comparison to traditional ICUs. Effective and efficient planning, along with appropriate allocation of resources, plays a crucial role in providing optimal care and improving patient outcomes. The significance of educating healthcare providers on the use of PUDs and ECG for bedside PICC placement is highlighted by the successful outcomes observed in this study. This success suggests that such techniques could be more widely utilized in other settings with limited resources or in the events of future infectious-disease outbreaks. Implementing management protocols can effectively reduce deviations from standard care practices. When performing invasive procedures on COVID-19 patients admitted to the ED, the ICU or the CDIU, it is essential to adhere to strict infection-control measures. This includes wearing full PPE, utilizing double gloving and using either an N95 respirator or a PAPR. These measures ensure safety of healthcare providers and minimize risk of transmission. PPE has become extremely common, and not everyone finds it comfortable to care for patients while wearing this new protective gear. Staff members have experienced various negative effects, such as dehydration and anxiety. The process of putting on and taking off PPE, which is crucial, also requires a considerable amount of time and energy. Shields and masks have affected communication between staff and patients, while the respirator/mask has hindered communication among staff members [14,15,16]. Because the CDIU in our institution was designed as a single-bed cell model, as opposed to in previous studies [17,18], this study was able to investigate effectiveness in a resource-limited situation. Due to the limited size of the CDIU, it was not feasible to install a C-arm fluoroscopy device, and using a portable radiography device would have required significant time and resources to maintain isolation. Additionally, it was nearly impossible to move patients to an angiography room or an operating room for the procedure to be performed by an interventional radiologist or a vascular surgeon under negative-pressure ventilation. Transferring infectious patients out of isolation rooms is often necessary, but it poses challenges in dealing with critically ill individuals due to the complexities involved in transporting medical equipment, ensuring proper disinfection and considering the patients’ clinical conditions. Therefore, using PUDs instead of vascular interventions is recommended to minimize the need for intrahospital patient transport [19,20,21]. The use of handheld ultrasonography in our clinic has increased considerably. Furthermore, conducting this procedure in the CDIU has the potential to alleviate patient anxiety and minimize the discomfort that arises from the need for transportation to other areas of the hospital. Additionally, exploring the potential cost-effectiveness of utilizing PUDs and ECG for PICC placement in CDIUs, as compared to traditional methods that demand greater resources and equipment, could be a valuable area of investigation in future research.

Earlier research has shown that PUDs in EDs have yielded comparable diagnostic results to those obtained through conventional high-end ultrasonography in dealing with patients who presented with sudden-onset undifferentiated dyspnea, chest pain and shock. Moreover, in the ICU, utilization of PUDs allows for expanded availability of focused assessment with sonography for trauma (FAST) examinations. Furthermore, it facilitates evaluation of free abdominal fluid or pleural effusion in the ICU setting. Additionally, PUDs enable visualization of pneumonic consolidations in patients with COVID-19 infections [21,22,23,24]. By minimizing the need for patient transport and in combination with PUDs, in our study, we employed ECG to reduce the risk of complications during bedside PICC insertion without fluoroscopic guidance. Our findings suggest that the incidence of catheter malpositioning was lower when ECG was used than when chest radiography was used. In a previous study [4], the overall proper positioning rate of the catheter tip upon the first attempt was 91.6%. This result suggests that PICC placement can be safely performed under PUD guidance and confirmed with ECG in facilities with limited resources, such as CDIUs. The occurrence of tip malposition following bedside PICC placement was rarely documented in the previous literature. However, Kim et al. [5] reported several risk factors associated with PICC malposition, including the female sex, advanced age, previous central catheter insertions, cancer and pulmonary diseases. Notably, all types of benign pulmonary diseases, such as acute respiratory distress syndrome, pneumonia, chronic obstructive pulmonary disease and bronchitis, were found to be linked with malposition. Pittiruti et al. recommended several following techniques in their commentary [6]: When selecting an insertion technique, it is recommended to prioritize the use of wireless ultrasound. These probes offer several advantages, including the ability to thoroughly clean the probe between patients, minimizing the risk of contamination. In cases where wireless ultrasound probes are not available, the best approach is to allocate an ultrasound device exclusively for use on COVID-19 patients; It is advised to avoid utilizing radiology procedures after cannulation due to the high risk of contamination for both operators and machinery. This includes avoiding transportation of patients to the radiology suite or bringing radiological equipment to a patient’s bed. To verify the location of the tip of the central venous catheter, non-radiological methods, such as transthoracic echocardiography and ECG, can be employed. The “bubble test,” involving infusion of saline with microbubbles of air and visualization through subxiphoid or apical echocardiography, can aid in determining tip location. Similarly, ECG can be easily performed at the bedside using either a standard ECG monitor or a dedicated wireless ECG monitor.

It is believed that use of PUDs to confirm placements of catheters in the internal jugular vein could further enhance the safety of PICC insertion. Although not performed in our study, previous research conducted by Schweickert et al. [25] demonstrated that bedside inspection of the ipsilateral neck, using ultrasound, can help identify malposition of the PICC tip. They reported that in the control group, out of the 149 attempted PICC insertions, 11 cases (7.4%) resulted in malposition of the catheter in the ipsilateral internal jugular vein. In contrast, in the intervention group where threaded PICCs were used, only one out of the one hundred and fifty-one insertions (0.7%) resulted in such malposition (*p* < 0.003). This approach enables successful repositioning of catheters during the initial procedure, thereby reducing the morbidity associated with bedside PICC placement. There are additional observations that are worthy of discussion. First, forty-eight (65.75%) patients underwent PICC insertion within 48 h of hospital admission, demonstrating that this procedure can be performed in CDIUs. Second, PICCs might be advantageous for patients with severe respiratory symptoms or for pronated patients. PICC placement may prevent pleuropulmonary complications or severe bleeding, which are associated with CVC insertion into the internal jugular vein [6]. Lastly, the right arm was accessed more frequently (*n* = 60, 82.19%) for two reasons: (1) the left arm was used for AVF creation or dialysis, and (2) it was more convenient for a right-handed operator to perform the procedure in the patient’s right arm, thus reducing the risk of complications [26].

Moreover, with regards to the pandemic and the evolving landscape of medical practices, there is a growing expectation with telemedicine. Use of ECG diagnosis for detecting myocardial infarctions (MIs) and arrhythmias has emerged as the prevailing approach in the field of telecardiology. While our investigation prioritized the significance of quarantine measures, we expect that further investigation and notable advancements in the procedure will do so for telemedicine and telemonitoring [27]. This study has a few limitations that should be considered. Firstly, it was conducted at a single center and was a retrospective study, which may limit the generalizability of the findings to other healthcare settings; further prospective and randomized studies must be assessed. Secondly, this study did not estimate the rate of upper-extremity DVT following insertion of PICCs after patients were discharged from the hospital. This is an important consideration, since this study focused on short-term outcomes and did not assess long-term results [28]. Notably, Chopra et al. [29] reported a high risk of DVT in critically ill patients with PICCs, which was up to 13.91%. Several explanations have been proposed to account for these observations. These include variations in the anatomical approach to the superior vena cava and a higher likelihood of mechanical trauma to the vessel intima in right-handed individuals. The discovery that PICCs inserted in the internal jugular vein, as opposed to arm veins, are linked to lower incidence of deep-vein thrombosis supports the theory that repeated arm movements, leading to intimal injury, may be associated with PICC-related deep-vein thrombosis. Additionally, the increased frequency of mechanical complications and longer dwell times of PICCs when compared to CVCs could potentially elevate the risk of deep-vein thrombosis. However, none of the patients in our study were diagnosed with DVT during their hospital stay. Consequently, our study reported a very low incidence of DVT. Numerous studies have highlighted occurrences of complications associated with hypercoagulability in COVID-19 patients. However, our study revealed that insertion of PICCs did not elevate the risk of upper-extremity DVT. Despite this finding, our study demonstrated the importance and feasibility of safely performing bedside PICC placement in CDIUs with limited resources during the pandemic. Although DVT is common, occurrence of pulmonary embolism was rare among individuals who underwent PICC placement [29]. As the potential for future outbreaks of infectious diseases persists beyond COVID-19, the demand for safe procedures that utilize scarce resources in isolation rooms remains a critical concern.

## 5. Conclusions

Bedside PICC placement using a PUD to guide vascular access and ECG to confirm catheter-tip location is a safe and feasible option for patients admitted to CDIUs.

## Figures and Tables

**Figure 1 jpm-13-00863-f001:**
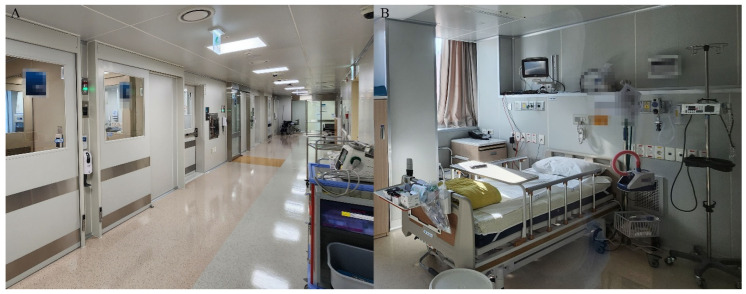
(**A**) Passageway of the communicable-disease isolation unit. (**B**) A single-cell room composition of the communicable-disease isolation unit.

**Figure 2 jpm-13-00863-f002:**
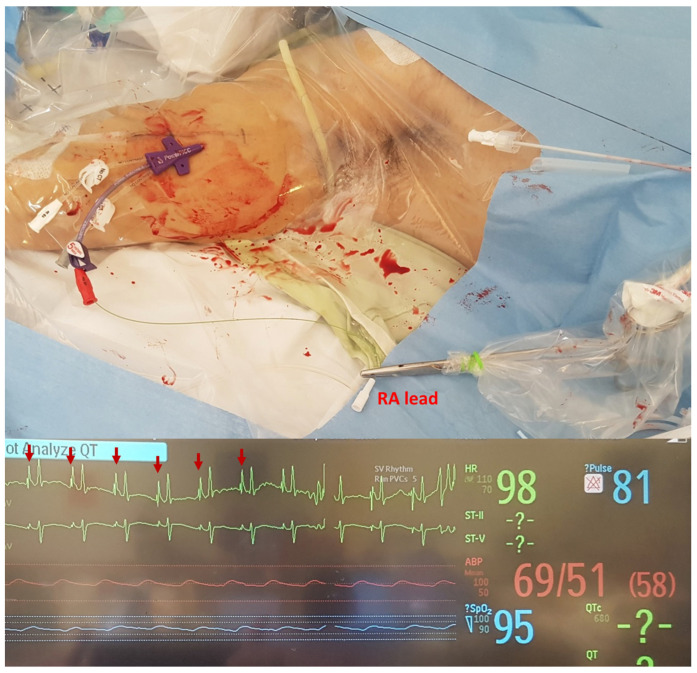
Waveform of continuous electrocardiographic monitoring. The red arrows indicate a giant P-wave.

**Table 1 jpm-13-00863-t001:** Clinical characteristics of the patients.

Characteristics	(*n* = 74)
Sex (Male, %)	40 (54.1)
Age (Years, Median (IQR))	69 (59–75)
Body Mass Index (kg/m^2^, Median (IQR))	24.96 (23.0–28.34)
Length of Hospital Stay (Days, Median (IQR))	26 (16–67)
Catheter Indwelling Time (Days, Median (IQR))	14 (9–22)
Access Side (Right, %)	61 (82.4)
Underlying Diseases	
Hypertension	39
Diabetes	35
Chronic Lung Disease	14
Chronic Heart Disease	12
Cancer	7
PICC Catheter Type	
PowerPICC™	45
Turbo-Ject^®^	29
Access Vein	
Basilic	57 (77.0)
Brachial	9 (12.1)
Cephalic	8 (10.8)
Interval Between CDIU Admission and PICC Insertion (Hours)	
<24	31 (41.9)
24–48	18 (24.3)
≥48	25 (33.8)
Confirmation of Catheter-Tip Location	
Electrocardiography	52 (70.3)
Chest radiography	21 (28.4)
Not Confirmed Immediately After the Procedure	1
Oxygen Therapy during the Procedure	
Room Air	3
Nasal Cannula	8
Mask with Reservoir	2
High-Flow Nasal Cannula	34
Mechanical Ventilation	27

PICC: peripherally inserted central catheter; CDIU: communicable-disease isolation unit.

## Data Availability

Data is unavailable due to ethical restrictions.

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
