# Peer review of "Feasibility of Ultrasound-Guided, Peripherally Inserted Central Catheter Placement at the Bedside in a Communicable-Disease Isolation Unit"

_jpm, 2023, doi:10.3390/jpm13050863_

Round 1
Reviewer 1 Report
I have reviewed the article entitled 'Feasibility of Ultrasound-Guided Peripherally Inserted Central Catheter Placement at Bedside in a Communicable Disease Isolation Unit'.
The article is interesting however several issues should be enlightened before further evaluation.
ECG is an interesting tool which is included in several risk scores and has a great prognostic value in addition to its role in this paper. Please add a short section about its role in the literature. Consider citing 'Machine Learning Approach on High Risk Treadmill Exercise Test to Predict Obstructive Coronary Artery Disease by using P, QRS, and T waves' Features' and 'A simple formula to predict echocardiographic diastolic dysfunction-electrocardiographic diastolic index'.
The pandemia also increases to the change in examinations of patients in addition to the changes in the procedures to the patients. Please add a short section about telemedicine. The great change in telemedicine in pandemia should be added to the discussion citing 'Telemedicine: Current Concepts and Future Perceptions'.
'
Author Response
RESPONSE TO REVIEWER 1:
Thank you for valuable opinion. As we accept your opinion we added following paragraph in the middle of the DISCUSSION.
"Moreover, with regards to the pandemic and the evolving landscape of medical practices, there is a growing expectation with telemedicine. The use of ECG diagnosis for detecting myocardial infarctions MI and ar-rhythmias has emerged as the prevailing approach in the field of telecardiology. While our investigation
prioritizes the significance of quarantine measures, we expect that further investigation and notable advance-ments in the procedure-telemedicine and telemonitoring [27]."
For this, a new reference [27] Hayıroğlu Mİ. Telemedicine: current concepts and future perceptions. Anatol J Cardiol. 2019 1;22(Suppl 2):21-2. was added.
On the other hand, citating following 'Machine Learning Approach on High Risk Treadmill Exercise Test to Predict Obstructive Coronary Artery Disease by using P, QRS, and T waves' Features' and 'A simple formula to predict echocardiographic diastolic dysfunction-electrocardiographic diastolic index' papers were out of scope of our investigation because our investigation was not focused on cardiology. Thank you for your understanding
Once again, Thank you for your time and effort for review our manuscript.
Reviewer 2 Report
Thank you for the opportunity to review this research article. The manuscript presents valuable findings and implications; however, there are some redundancies and areas that need further attention. Below is a summary of my comments and recommendations.
Strengths
1. The study's main findings and their implications are well-presented, contributing to the understanding of ultrasound-guided PICC placement in CDIUs.
Areas for Improvement:
1. Some redundancy in the presentation of information in the abstract needs to be addressed for clarity.
2. The introduction, methods, and discussion sections require elaboration and clarification on specific points.
Specific Comments:
Abstract:
- The abstract highlights the main findings and their implications; however, some redundancy in the presentation of information should be addressed for clarity.
Introduction:
- The introduction lacks a brief explanation of the ultrasound-guided technique for PICC placement, which is the focus of the study.
- There should be a comparison of ultrasound-guided PICC placement with other techniques to establish why ultrasound guidance is the technique of interest in this study.
- The challenges associated with performing bedside PICC placement in CDIUs, such as limited resources and higher patient-to-nurse ratios, should be addressed.
Methods:
- The methods provide a detailed description of the study design, population, and setting; however, the primary and secondary endpoints should be stated more clearly and concisely.
Discussion:
- The discussion does not mention the following potential implications:
1. Training and skill development: The successful use of PUD and ECG for bedside PICC placement in this study highlights the importance of training healthcare providers in these techniques. This may lead to broader adoption in other resource-limited settings or during future infectious disease outbreaks.
2. Cost-effectiveness: Using PUD and ECG for PICC placement in CDIUs could potentially be more cost-effective than traditional methods requiring more resources and equipment. This aspect could be further explored in future research.
3. Patient comfort and satisfaction: The potential impact of bedside PICC placement on patient comfort and satisfaction should be discussed. Performing the procedure in the CDIU may help reduce patient anxiety and discomfort associated with transportation to other hospital areas.
4. Impact on hospital workflow: The use of PUD and ECG for bedside PICC placement in CDIUs may have implications for hospital workflow, potentially reducing the strain on radiology and vascular surgery departments during infectious disease outbreaks.
5. Infection control implications: The study should discuss the potential implications of bedside PICC placement on infection control. Minimizing patient transport and using less invasive techniques may help reduce the risk of disease transmission among healthcare providers and patients.
In conclusion, the manuscript has potential, but certain aspects need further elaboration and clarification. Addressing these comments will improve the clarity and impact of the study. I look forward to reviewing a revised version of the manuscript.
The quality of the English language requires significant improvement.
Author Response
Response to Reviewer 2 Comments
1. Abstract: - The abstract highlights the main findings and their implications; however, some redundancy in the presentation of information should be addressed for clarity.
Answer: Thank you for your valuable comment: we make the abstract more concisely as following.
"Background: Previous studies have investigated the safety of peripherally inserted central catheters (PICCs) in the intensive care unit (ICU). However, it remains uncertain whether PICC placement can be successfully carried out in settings with limited resources and a challenging environment for procedures, such as com-municable disease isolation units (CDIUs). Methods: This study investigated the safety of PICC in patients admitted to CDIU. The researchers used a handheld portable ultrasound device (PUD) to guide venous ac-cess and confirmed the catheter tip location with electrocardiography (ECG) or portable chest radiog-raphy.Results: Out of 74 patients, the basilic vein and the right arm were the most common access site and location, respectively. The incidence of malposition was significantly higher with chest radiography com-pared to ECG. (52.4% vs. 2.0%, p < 0.001). Conclusions: Using a handheld PUD to place PICCs at the bed-side and confirming the tip location with ECG is a feasible option for CDIU patients."
2. Introduction:
2-1. The introduction lacks a brief explanation of the ultrasound-guided technique for PICC placement, which is the focus of the study.
2-2. There should be a comparison of ultrasound-guided PICC placement with other techniques to establish why ultrasound guidance is the technique of interest in this study.
Answer 2-1 and 2-2: we additionally explained with following sentence "The supe-riority of the current technique is evident when compared to the previous blind method or traditional PICC placement which performed by an interventional ra-diologist in the interventional radiology suite under fluoroscopic guidance."
2-3. The challenges associated with performing bedside PICC placement in CDIUs, such as limited resources and higher patient-to-nurse ratios, should be addressed.
Answer 2-3: We suggested nurse-to-bed ratio and nurse-to-patient ratio with reference [7] Fan, E.M.P.; Nguyen, N.H.L.; Ang, S.Y.; Aloweni, F.; Goh, H.Q.I.; Quek, L.T.; Ayre, T.C.; Pourghaderi, A.R.; Lam, S.W.; Ong, E.H.M. Impact of COVID‐19 on acute isolation bed capacity and nursing workforce requirements: A retrospective review. Journal of nursing management 2021, 29, 1220-1227. and [8]. Lilly, C.M.; Oropello, J.M.; Pastores, S.M.; Coopersmith, C.M.; Khan, R.A.; Sessler, C.; Christman, J.W. Workforce, workload, and burnout in critical care organizations: survey results and research agenda. Critical care medicine 2020, 48, 1565.
3. Methods:
3-1. The methods provide a detailed description of the study design, population, and setting; however, the primary and secondary endpoints should be stated more clearly and concisely.
Answer 3: As your comment, we state more clearly: "The primary endpoint involved comparing the number of patients in whom catheter tip position was
verified using ECG versus those confirmed using chest radiography. The secondary endpoint was the occur-rence of catheter malposition when confirming the catheter tip position using both ECG and chest radiog-raphy methods."
4. Discussion:
4-1. Training and skill development: The successful use of PUD and ECG for bedside PICC placement in this study highlights the importance of training healthcare providers in these techniques. This may lead to broader adoption in other resource-limited settings or during future infectious disease outbreaks.
Answer 4-1: Commented in discussion as following: "The significance of educating healthcare providers in the use of PUD and ECG for bedside PICC placement is highlighted by the successful outcomes observed in this study. This
success suggests that such techniques could be more widely utilized in other settings with limited resources or in the event of future infectious disease outbreaks."
4-2. Cost-effectiveness: Using PUD and ECG for PICC placement in CDIUs could potentially be more cost-effective than traditional methods requiring more resources and equipment. This aspect could be further explored in future research.
4-3. Patient comfort and satisfaction: The potential impact of bedside PICC placement on patient comfort and satisfaction should be discussed. Performing the procedure in the CDIU may help reduce patient anxiety and discomfort associated with transportation to other hospital areas.
Answer 4-2 and 4-3: Described as following "Furthermore, conducting the procedure in the CDIU has the potential to alleviate patient anxiety and minimize discomfort that arises from the need for transportation to other areas of the hospital. And exploring the potential cost-effectiveness of utilizing PUD and ECG for PICC place-ment in CDIUs, as compared to traditional methods that demand greater resources and equipment, could be a valuable area of investigation in future research.""
4-4. Impact on hospital workflow: The use of PUD and ECG for bedside PICC placement in CDIUs may have implications for hospital workflow, potentially reducing the strain on radiology and vascular surgery departments during infectious disease outbreaks.
4-5. Infection control implications: The study should discuss the potential implications of bedside PICC placement on infection control. Minimizing patient transport and using less invasive techniques may help reduce the risk of disease transmission among healthcare providers and patients.
Answer 4-4 and 4-5: Commented by two sentenses as following: Additionally, it was nearly impossible to move patients to an angiography room or operating room for the procedure to be performed by an intervention-radiologist or vascular surgeon under negative-pressure ventilation." and " By minimizing the need for patient transport and in combination with PUD, in our study, we employed ECG to reduce the risk of complications during bedside PICC insertion without fluoroscopic guidance."
All changes are yellow-highlited in maintext.
Once again, thank you for your time and effort for reading and reviewing our manuscript.
Reviewer 3 Report
This study of Won Yoon et al. describes de rate of malposition of PICCs in a special ICU with limited resources for the admission of COVID patients comparing those PICCs inserted with portable ultrasonography versus portable chest Rx devices. They found that ECG was safer and feasible to confirm PICC tip location, as malposition was reduced from 52.4% to 2.0% compared to Rx.
In my opinion I would convert the article to a short letter by reducing discussion. The main limitation of the study is its retrospective nature.
Other minor comments to addressed:
1. Put all percentage values with only one decimal.
2. I will remove figure 2, as it doesn’t provide additional information to that detailed in the text referring to continuous ECG monitoring. I would rather add a photo of a PICC insertion using ECG.
Table 1: Detail in each variable how data are presented (median (IQR) or N (%)) instead of in the table legend.
4. Did the authors make a comparison of clinical variables between those patients who underwent through ECG vs. Rx?
5. Add in the limitation section that it was a retrospective study, and further prospective and randomized studies must be assessed.
Author Response
Response to Reviewer 3 Comments
1. Put all percentage values with only one decimal.
Answer 1: we changed table percetage values to only decimal values. Thank you for your comment.
2. I will remove figure 2, as it doesn’t provide additional information to that detailed in the text referring to continuous ECG monitoring. I would rather add a photo of a PICC insertion using ECG.
Answer 2: As your recommendation, we changed photo(figure 2).
3. Table 1: Detail in each variable how data are presented (median (IQR) or N (%)) instead of in the table legend.
Answer 3: We have changed as your recommendation.
4. Did the authors make a comparison of clinical variables between those patients who underwent through ECG vs. Rx?
Answer 4: in result section, we commented as following "Out of these, a total of 12 cases of malposition were reported, with 11 occurring in the radiography group and 1 in the ECG group. Immediate re-insertion of the catheter was performed in these cases, which accounted for 52.38% in the radiography group compared to 1.96% in the ECG group (p < 0.001). It is important to interpret this result cautiously, as radiography was used as a sequential method following PICC placement when ECG was not suitable, particularly in patients with arrhythmia. "
5. Add in the limitation section that it was a retrospective study, and further prospective and randomized studies must be assessed.
Answer 5. We totally agree on your opinion: we changed as following: "Firstly, it was conducted at a single center and retrospective study, which may limit the generalizability of the findings to other healthcare settings and further prospective and randomized studies must be assessed. "
All changes were yellow-highlighted in maintext.
Once again, thank you for your time and effort for reading our manuscript.
Round 2
Reviewer 1 Report
Thank you for the required revisions.